# Effect of Honey and Syrup Diets Enriched with 1,3-1,6 β-Glucans on Honeybee Survival Rate and Phenoloxidase Activity (*Apis mellifera* L. 1758)

**DOI:** 10.3390/vetsci8070130

**Published:** 2021-07-13

**Authors:** Simona Sagona, Baldassare Fronte, Francesca Coppola, Elena Tafi, Matteo Giusti, Lionella Palego, Laura Betti, Gino Giannaccini, Lorenzo Guglielminetti, Antonio Felicioli

**Affiliations:** 1Department of Veterinary Sciences, University of Pisa, viale delle Piagge 2, 56124 Pisa, Italy; simona.sagona@unipi.it (S.S.); baldassare.fronte@unipi.it (B.F.); francesca.coppola@vet.unipi.it (F.C.); elena.tafi@unibas.it (E.T.); giusti.matteo@hotmail.it (M.G.); 2Department of Pharmacy, University of Pisa, Via Bonanno 6, 56126 Pisa, Italy; laura.betti@unipi.it (L.B.); gino.giannaccini@unipi.it (G.G.); 3Department of Science, University of the Study of Basilicata, Via dell’Ateneo Lucano 10, 85100 Potenza, Italy; 4Department of Clinical and Experimental Medicine, University of Pisa, Via Savi 10, 56126 Pisa, Italy; lionella.palego@unipi.it; 5Department of Agriculture, Food and Environment, University of Pisa, Via del Borghetto 80, 56124 Pisa, Italy; lorenzo.guglielminetti@unipi.it

**Keywords:** *Apis mellifera*, β-glucans, innate immune system, nutrition, survival, phenoloxidase activity

## Abstract

β-glucans can activate the animal innate immune system by acting as immune-modulators and inducing various stimulatory effects. The aim of this study was to investigate the effect of 1,3-1,6 β-glucans administered orally for 96 h on *Apis mellifera* workers (newly emerged and nurse bees). β-glucans were included in honey and syrup. Survival rate and phenoloxidase activity were measured. In both newly emerged and nurse bees, β-glucans supplementation did not affect survival rate (*p* > 0.05). Conversely, phenoloxidase activity was higher in both newly emerged bees (*p* = 0.048) and nurse bees (*p* = 0.014) fed with a honey diet enriched with β-glucans compared to those fed with only honey. In both the newly emerged and nurse bees, no statistical differences in phenoloxidase activity were recorded between the group fed with a syrup-based diet enriched with β-glucans and the control group (*p* > 0.05). The absence of significant variation in survival suggests that the potential negative effect of β-glucans in healthy bees could be mitigated by their metabolism. Conversely, the inclusion of β-glucans in a honey-based diet determined an increase of phenoloxidase activity, suggesting that the effect of β-glucan inclusion in the diet of healthy bees on phenoloxidase activity could be linked to the type of base-diet. Further investigations on β-glucans metabolism in bees, on molecular mechanism of phenoloxidase activation by 1,3-1,6 β-glucans, and relative thresholds are desirable. Moreover, investigation on the combined action of honey and β-glucans on phenoloxidase activity are needed.

## 1. Introduction

Chronic bee life-span reduction occurring mainly in the winter is a key problem in beekeeping that has not yet been completely solved. No suitable and legal drugs that work against bee pathogens are yet available. For this reason, the administration of lactic acid bacteria [1], essential oils [2], and supplement diets [3,4,5,6,7] were tested in order to elicit information regarding the immune systems of honeybees, minimize the occurrence of toxic residues in hive products, and prevent the development of resistance phenomena to drugs. A honeybee colony is a super-organism characterized by two levels of innate immunity: a social-scale immunity and an individual-scale immunity [8,9]. The social-scale innate immunity is described as a group of hygienic behaviours, including the secretion of glucose oxidase, which is an enzyme involved in the production of antimicrobial hydrogen peroxide and gluconic acid from glucose [10,11,12]. Glucose oxidase activity is higher in nurse bees and in young foragers than in newly emerged bees and old foragers [11]. The individual-scale innate immunity consists of cellular and molecular specific mechanisms, such as phagocytosis, antimicrobial peptides, lectins, complement-like factors [13], and the pro-phenoloxidase cascade [14]. The pro-phenoloxidase is regulated (activation or inhibition) by specific proteinases or proteinases inhibitors and by the presence of small amounts of microbial-origin compounds such as 1,3 β-glucans, lipopolysaccharides, and peptidoglycans [15]. Excess phenoloxidase activity induces the presence of several potentially dangerous highly reactive quinone intermediates [15] and an inhibition system is activated to control the immune self-stimulation and to avoid a possible excess of these metabolites [16,17]. In honeybees, phenoloxidase activity increases with the age of worker bees and reaches a plateau within the first week of adult life [18].

β-glucans are polysaccharides that have the ability to stimulate the immune system, thus behaving as immunomodulators [19]. The ability of β-glucans to activate the innate immune system of vertebrates and invertebrates has been described by several authors [19,20,21,22,23]. β-glucans can induce all of the major antimicrobial immune mechanisms found in invertebrates, including humoral (coagulation and anti-microbial peptide production), cellular, and phenoloxidase responses [19]. Richard et al. [24] confirmed immune system activation in *Apis mellifera* by measuring expression levels of the *Defensin2* immune response gene, and by monitoring the social immune effects after injection of a solution containing lipopolysaccharides (LPS), molecules that activate the immune system in a similar way to β-glucans. In a detailed investigation of behaviour (a part of social immunity), Richard et al. [24] observed that LPS-injected subjects received more non-agonistic social contacts (antennating and allogrooming) than control ones. Mowlds et al. [25] observed a potentiated immune response in *Galleria mellonella* larvae after inoculation with high doses of β-glucans. Furthermore, Mazzei et al. [23] and Felicioli et al. [7] suggested that β-glucans might contribute to increased honeybee resistance to viral infection by observing the effects of oral administration of this natural molecule in a challenge test with deformed wing virus (DWV) on newly emerged bees. The aim of this study was to investigate the effect of the inclusion of β-glucans in honey- and syrup-based diets in apparently healthy newly emerged and nurse bees after 96 h of feeding. We predicted that the inclusion of β-glucans in the diets of healthy newly emerged and nurse bees would determine a reduction of survival and an increase of phenoloxidase activity.

## 2. Materials and Methods

### 2.1. Sampling and Honeybees Rearing

Newly emerged bees and nurse bees were chosen for the study, since Schmid and colleagues [18] observed that phenoloxidase activity changes accordingly with the age of bees, thus being higher in nurse bees compared to newly emerged bees.

Newly emerged bees and nurse bees were collected in 2015 from five hives in the experimental apiary of the Department of Veterinary Sciences of Pisa University (latitude 43°40′51.45″ N; longitude 10°20′50.96″ E). Each hive was previously managed in order to achieve the same family strength (adult/brood ratio) and queen age (2 years old). In all hives the same treatment against *Varroa* mite was performed and no symptoms attributable to the principal honeybee diseases (American foulbrood, deformed wings, and diarrhoea) were detected. Newly emerged bees were collected in spring-summer from frames containing broods by use of tweezers as they emerged from their cells. Nurse bees were selected by observing their typical brood food provisioning behaviour during spring-summer. 

The collected bees were reared for 96 h in glass jars and fed with control and experimental diets throughout the investigation period, at a temperature of 30 ± 2 °C and a relative humidity of 60%. Each jar of 750 cm^3^ (h 15 cm; ∅ 8 cm) was laid on its side and equipped with a metal cap with two holes: a 1.5 cm hole for the syringe used as the feeder, and another smaller hole for a 2 mL syringe used as a water dispenser. Food and water were administered ad libitum to honeybees and renewed daily.

Moreover, five supplementary small holes for air were added. The jar was equipped with a piece of comb (about 5 cm × 8 cm) taken from the experimental apiary (from the same colony and free of brood, pollen, or beebread) and a sheet of white paper, to reduce the stress of cage rearing. The bees were distributed randomly into the jars after collection, in order to have an equal representation from the source colonies in the jars. A four day duration of experimental feeding was chosen in order to test if the effects of β-glucan were compatible with a reasonable time frame for beekeepers to feed their bees.

Newly emerged bees and nurse bees were fed four different diets (honey, H0; honey and 0.5% (*w*/*w*) β-glucans, H0.5; syrup (sugar-water), S0; syrup and 0.5% (*w*/*w*) β-glucans, S0.5). The supplementation of the honeybees’ diet with 1,3-1,6 β-glucans was performed using the commercial product MacroGard^®^ (Biorigin^©^, Lençóis Paulista, Brazil) containing β-glucans extracted from *Saccharomyces cerevisiae* cell walls. MacroGard has a content of 1,3-1,6 β-glucans > 60%. The sugar water syrup was a fresh mixture composed of 38% (*w*/*v*) fructose, 32% (*w*/*v*) glucose, and 15% (*w*/*v*) sucrose. The syrup diet was chosen to test the effect of 1,3-1,6 β-glucan as a supplementary food usually used in beekeeping. The honey diet was chosen in order to preliminarily investigate the effect of 1,3-1,6 β-glucan as a supplementary diet in beekeeping in presence of natural bee food. A concentration of 0.5% (*w*/*w*) of 1,3-1,6 β-glucan was chosen in accordance with Mazzei et al. [23] who reported an increase in honeybee survival rate and phenoloxidase activity after 13 days of experimental feeding using this β-glucan dosage. The β-glucans were weighed according to the weight percentage chosen and mixed with the diet (honey or syrup) using a magnetic stirrer. For each control and experimental diet provided to newly emerged and nurse bees three replicates were performed. Each replicate was a jar containing 25 bees. 

### 2.2. Melissopalynological Analysis

Honey used in the experimental trials was provided by a local beekeeper. The composition of honey was assessed by melissopalynological analysis. For the melissopalynological analysis, 40 mL of water were added to 10 g of honey. After two centrifuges at 3000 rpm for 10 min, the pellet was collected, dried, and fixed on a microscope slide using glycerol jelly [26]. 

Melissopalynological identification was performed by optical microscopy with total magnification (400× and 1000×). A reference collection of pollen of Pisa University (Italy) and different guides on pollen morphology were used for the recognition of the pollen types [27,28,29,30,31]. Melissopalynological profiles were compared with guidelines by Colombo et al. [32], in order to confirm the botanical origins of the honey samples. The nomenclature used was in accordance with Oddo and D’Albore [33].

### 2.3. Survival Rate and Phenoloxidase Activity (PO)

Survival of bees was recorded once a day throughout the investigation period and the dead bees were removed from the jars every day. After 96 h of rearing had passed, all the surviving honeybees were stored at −20 °C until enzymatic assay. The data of the PO investigation in newly emerged honeybees came from three replicates for diet (*n* = 18 per diet), as did the data of the nurse honeybees (*n* = 36 per diet).

To determine phenoloxidase activity, a single bee head was weighted and mixed with the extraction buffer (50 mM phosphate buffer pH 7.2, 1% Triton X-100), and then homogenized with a Teflon pestle. After a centrifuge at 4000 rpm at 4 °C for 15 min, the supernatant was collected and stored at −20 °C until PO assay. PO assay was based on the Alaux protocol [3] with some modifications [23]. Fifty µL of protein extract was added to a solution composed of 41.3 mM phosphate saline buffer pH 7.4 and incubated at 37 °C for 5 min. Afterwards, L-3,4-dihydroxyphenylalanine (L-dopa) (2 mg/mL), as substrate, was added and the absorbance data were recorded at λ = 490 nm at 0 and 10 min by an Ultrospec 2100 UV pro spectrophotometer (Amersham Biosciences, Little Chalfont, UK). Results are expressed as mU/min/mg of tissue (as mean and standard deviation). The calibration curve was performed by different melanin concentrations using the same protocol described above. The first concentration of melanin was 1 mg/mL of melanin in 30% ammonium hydroxide. The other concentrations were obtained by diluting the first solution in milliQ water.

### 2.4. Statistical Analysis

A statistical analysis was performed using JMP software (SAS Institute, Cary, NC, USA 2008) [34]. Survival rate data were obtained as the sum of the surviving bees in all replicates for each diet. For each age group of honeybees, the survival rate data of all diet groups were simultaneously analysed by Log-rank test using the product-limit (Kaplan–Meier) method for factors of right-censored data. The PO analysis was tested for normal distribution by means of the Shapiro–Wilk test. Since the data results were not normally distributed, the replicates for each treatment were tested by the Kruskal–Wallis H-test. Since no significant differences were observed among the three replicates for each diet, the replicates within the treatments were pooled. The difference between H0 vs. H0.5 or S0 vs. S0.5 was assessed by using the non-parametric Wilcoxon test. Differences were considered significant if associated with a *p* value < 0.05.

## 3. Results

### 3.1. Melissopalynological Analysis 

The results of the melissopalynological analysis showed that the honey administered in the bees diets was composed of 18% *Robinia* pollen and attributable to Acacia honey (Table 1).

### 3.2. Survival Rate

#### 3.2.1. Newly Emerged Bees Survival 

The survival rate of the newly emerged honeybees fed different base-diets (syrup and honey) enriched with β-glucans (0.5% *w*/*w*) are reported in Figure 1.

No differences were observed between the newly emerged bees fed a honey or syrup base-diet and those fed diets supplemented with β-glucans (0.5% *w*/*w*), *p* = 0.750 Log-rank χ^2^_1_ = 0.102 and *p* = 0.147 Log-rank χ^2^_1_ = 2.099, respectively.

#### 3.2.2. Nurse Bees Survival

The survival rate of the nurse bees fed with β-glucans supplementation (0.5 % *w*/*w*) in the different base-diets (honey and syrup) and control groups are reported in Figure 2. 

No differences were observed between the nurse bees fed honey or syrup base-diets and those with β-glucans-enriched diets (0.5% *w*/*w*), *p* = 0.377 Log-rank χ^2^_1_ = 0.780 and *p* = 0.507 Log-rank χ^2^_1_ = 0.440, respectively.

### 3.3. Phenoloxidase Activity

#### 3.3.1. Phenoloxidase Activity in Newly Emerged Bees

Phenoloxidase activity in newly emerged honeybees fed different base-diets (syrup and honey) with added β-glucans (0.5% *w*/*w*) and control groups are reported in Figure 3.

Within the honey-based diet group, newly emerged bees fed a honey diet enriched with β-glucans showed higher phenoloxidase activity compared to the control group (χ^2^_1_ = 3.891, *p* = 0.0485). No statistical differences were recorded between the group fed a syrup-based diet enriched with β-glucans and the control group (χ^2^_1_ = 0.393, *p* = 0.5307).

#### 3.3.2. Phenoloxidase Activity in Nurse Bees

The phenoloxidase activity observed in nurse bees fed different experimental diets and the control groups are reported in Figure 4.

The nurse bees fed a honey diet enriched with β-glucans showed higher phenoloxidase activity compared to the control group, (χ^2^_1_ = 5.974, *p* = 0.0145). No statistical differences were recorded between the group fed a syrup-based diet enriched with β-glucans and the control group (χ^2^_1_ = 1.385, *p* = 0.239).

## 4. Discussion

The effects of 1,3-1,6 β-glucans supplementation over a short period (96 h) in apparently healthy honeybees diet were here investigated. The results obtained in this investigation indicate that inclusion of β-glucans in the diets of healthy bees did not affect their survival. This result contradicts the prediction that the supplementation of β-glucans would determine a reduction in the survival rate of the bees. However, Felicioli and colleagues [7] observed a higher survival rate in newly emerged bees fed only on syrup than those fed with syrup enriched with 0.5% β-glucans in a 24-day experimental trial. In detail, Felicioli and colleagues [7] investigated the effects of β-glucans administration for 24 days on naturally DWV-infected newly emerged bees. The negative effects of β-glucans supplementation in the syrup-based diet of bees recorded by Felicioli and colleagues [7], and not present in the current investigation, could be due to the synergic negative effect of β-glucans with DWV pathology. At the same time, it could be hypothesized that the potential negative effect of β-glucans in healthy bees could be mitigated by their metabolism. Further investigations on β-glucans metabolism in bees are desirable.

Moreover, the nurse bees collected in this investigation were already 6–12 days old and had already consumed honey and pollen before being taken from the family for the experimental trial. Therefore, the absence of differences recorded in this investigation in survival between the experimental and control groups may be due to the absence of nutritional stress following diet protein exclusion for a short time. However, it cannot be ruled out that the intake of β-glucans over a longer period of time may influence nurse bees’ survival both positively and negatively. Winter honeybees have a longer physiological life than honeybees in spring and summer [35], which might suggest a greater resistance of winter bees to stress factors. Dainat and colleagues [36] suggest that honeybee mortality in winter could be associated to the replication of DWV in bee tissues. However, according to Felicioli et al. [7] and Mazzei et al. [23], β-glucans might contribute to increasing honeybee resistance to DWV infection. Therefore, it would be interesting to investigate the survival rate in winter bees fed with diets enriched with β-glucans.

In this investigation, both newly emerged and nurse bees fed honey-based diets supplemented with 0.5% β-glucan showed an increase of phenoloxidase activity compared to those fed only with honey. Conversely, no differences were recorded in the phenoloxidase activity in both newly emerged and nurse bees fed syrup-based diets. No differences in phenoloxidase activity were also recorded by Felicioli and colleagues [7] between newly emerged bees fed a 0.5% β-glucans enriched syrup-diet and newly emerged fed only syrup on day 24 of feeding. Therefore, the results obtained in this investigation suggest that the effect of supplements β-glucans into the diets of healthy bees on phenoloxidase activity could be linked to the type of base-diet. It could be speculated that a component of honey could act on the metabolism of β-glucans or give an additive effect to phenoloxidase activity. Further investigations are needed to clarify the combined action of honey and β-glucans on phenoloxidase activity.

Moreover, since pollen contains β-glucans and the natural diet of young bees is based on honey and pollen [37,38], it could also be interesting to investigate the survival and phenoloxidase activity in bees fed honey, pollen, and β-glucans.

## 5. Conclusions

The dietary administration of 0.5% (*w*/*w*) 1,3-1,6 β-glucans for 96 h of feeding did not affect newly emerged and nurse bees’ survival rates, but did stimulate the activation of phenoloxidase activity in the bees fed a honey base-diet. These results could confirm the ability of β-glucans to affect honeybee immune system. Moreover, although insects do not have memory cells, Sadd and Schmid-Hempel [39] observed a prompt immune response to repeated pathogen exposition. Therefore, the administration of 1,3-1,6 β-glucans could be used in order to “prepare” bees to possible pathogen exposition, without modifying their lifespan. Further investigation on the molecular mechanisms underlying phenoloxidase activation by 1,3-1,6 β-glucan, together with the appraisal of their relative threshold levels, are desirable.

## Figures and Tables

**Figure 1 vetsci-08-00130-f001:**
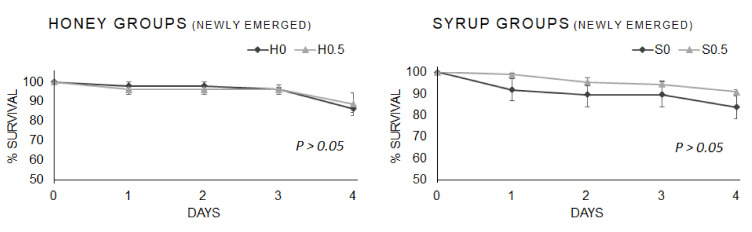
Survival rate of newly emerged bees fed syrup or honey base-diets, supplemented with β-glucans (0.5% *w*/*w*), and control groups. Each curve indicates the sum of three replicates per group. The bars indicate the standard error calculated on the survival of the three replicates per each group. H = honey; S = syrup; 0, 0.5 = % (*w*/*w*) 1,3-1,6 β-glucans.

**Figure 2 vetsci-08-00130-f002:**
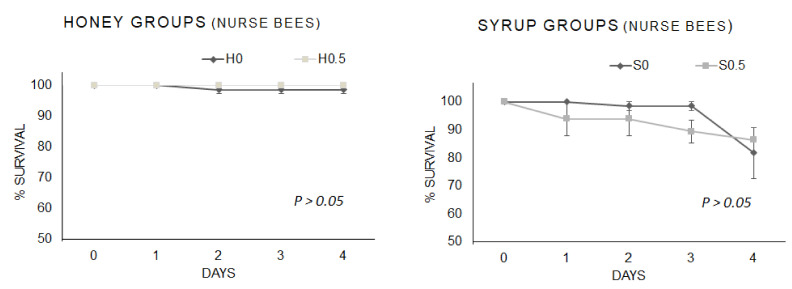
Survival rate of nurse bees fed base-diets (honey and syrup) enriched with β-glucans (0.5 % *w*/*w*) and control groups. Each curve indicates the sum of three replicates per group. The bars indicate the standard error calculated on the survival of the three replicates per each group. H = honey; S = syrup; 0, 0.5 = % (*w*/*w*) 1,3-1,6 β-glucans.

**Figure 3 vetsci-08-00130-f003:**
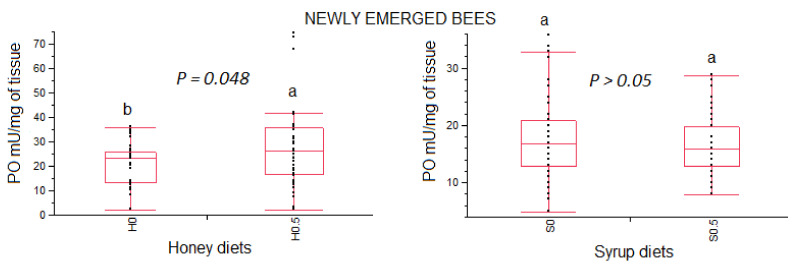
Box plots of the phenoloxidase activity (PO) in newly emerged bees fed different base-diets (honey and syrup) with added β-glucans (0.5% *w*/*w*). Data are presented as medians and range (min and max values). Different lowercase letters above the bars indicate statistical differences between honey or syrup diets according to the non-parametric Wilcoxon test. H = honey; S = syrup; 0, 0.5 = % (*w*/*w*) 1,3-1,6 β-glucans.

**Figure 4 vetsci-08-00130-f004:**
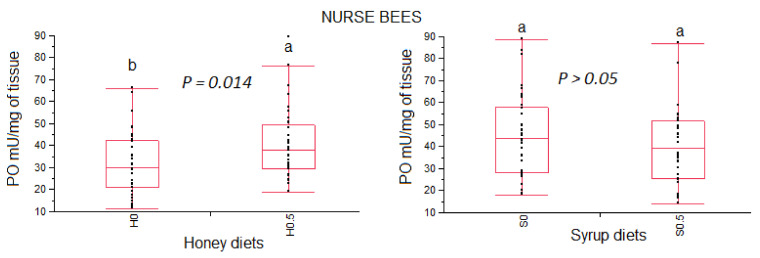
Box plots representing the phenoloxidase activity (PO mU/mg tissue) in nurse bees fed different diets. Data are presented as medians and range (min and max values). Different lowercase letters above the bars indicate statistical differences between honey or syrup diets according to the non-parametric Wilcoxon test. H = honey; S = syrup; 0, 0.5 = % (*w*/*w*) 1,3-1,6 β-glucans.

**Table 1 vetsci-08-00130-t001:** Melissopalynological analysis of the honey used in the experimental trials.

Pollen Species	Pollen Presence (%)	Pollen Species	Pollen Presence (%)
*Acer pseudoplatanus gr*	P	*Palmae* (*Chamaerops f*)	P
*Actinidia*	1	*Potentilla*/*Fragaria f*	13
*Asparagus acutifolius gr*	P	*Prunus f*	1
*Brassica f*	1	*Quercus ilex gr*	25
*Castanea*	3	*Ranunculus repens gr*	P
*Eucalyptus*	P	*Robinia*	18
*Fraxinus ornus*	11	*Rubus f*	1
*Gleditsia*	1	*Salix*	3
*Graminaceae*	2	*Sambucus nigra*	4
*Hedera*	1	*Sedum f*	4
*Juncus*	P	*Trifolium repens gr*	1
*Lotus*	P	*Urticaceae*/*Moraceae*	1
*Mercurialis*	P	*Verbascum*	3
*Olea f*	2		

Note: gr = group; f = form; P = pollen presence but with a percentage lower than 1.

## Data Availability

All data are available from the corresponding author.

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
