# Peer review of "Effect of Honey and Syrup Diets Enriched with 1,3-1,6 β-Glucans on Honeybee Survival Rate and Phenoloxidase Activity (Apis mellifera L. 1758)"

_vetsci, 2021, doi:10.3390/vetsci8070130_

Round 1

Reviewer 1 Report

This manuscript by Sagona and the colleague have the effect of honey and syrup diets enriched with 1,3-1,6 β-glucans on honeybee wellness. It is an interesting topic. However, the writing is very poor. The following revision could improve the quality of the paper.

  1. Line 16, Please correct ‘Aim of this study’ to ‘The aim of this study’.
  2. Abstract should not separated as two paragraph.
  3. Lines 19-23, please described the degree of the changes. Meanwhile, separate the results and explanation of the changes.
  4. Abstract, please added the P value for the significant changes.
  5. Lines 50-51, only two rows should not separated as a paragraph. This is not the right way to write the scientific papers.
  6. Lines 80-82, similar to the previously, only three rows should not separated as a paragraph.
  7. Lines 105-106. similar to the previously, only two rows should not separated as a paragraph. Please check throughout the papers.
  8. Line 161, ‘Each curve indicates the sum of 3 replicates per group.’, please added the SD or SE for the values. Meanwhile, only 3 replicates per group seems not enough for the statistical analysis.
  9. Line 164, ‘P>0.05’ correct to ‘P > 0.05’please check throughout the paper.
  10. Line 171, , ‘Each curve indicates the sum of 3 replicates per group.’, please added the SD or SE for the values. Meanwhile, only 3 replicates per group seems not enough for the statistical analysis.
  11. Line 174, ‘P>0.05’ correct to ‘P > 0.05’please check throughout the paper.
  12. Figures 4 and 5, please explain the P value and the letters used in the figures.

Reviewer 2 Report

The following comments are offered:  

Line 60: Please include more detail on the social immune effects.  These studies demonstrated that social immune effects were modulated by B-glucans?

Line 80-81: How were the bees sourced?  Did all cages/treatments have equal representation from the source colonies?

Line 88-91: Likewise for the comb section... Did all cages/treatments receive a section of comb from the same colony?  If not, were comb sections devoid of pollen/beebread?  

Line 91: Please justify beyond beekeeper feeding, why the survival study was limited to 96 hours? The issue comes up in the Discussion as well (Lines 217-218). Presumably, a beekeeper would apply feed, return a few days later, and remove/replenish feed, but the bees would either store this feed or consume it directly; therefore, their exposure would be immediate or potentially, the effects would be longer than 4 days.  Cage studies typically run until the majority/all of bees in a treatment/study succumb.  The exposure to a treatment may be brief (acute) or chronic, but the monitoring of survival is sustained. This is apparent in Figure 2.  If the study was taken out longer, it may have shown that syrup bees have a greater rate of mortality than syrup bees given B-glucans. Unless the test compound is acutely/highly toxic, effects of exposure on survival are often not observed within the first 7-10 days. 

Line 96: MacroGard only contains B-glucans? This is important, please include the list of materials and their amounts contained within the MacroGard product.  From what source material and how are B-glucans extracted?  Why was 0.5% w/w chosen as the exposure dose?  Was a preliminary study conducted with different concentrations?  This needs to be justified. 

Line 107-118: Unclear the importance of including the palynological analysis. Was this to rule out confounding effects (e.g., health benefits?) of pollen or phytochemicals in honey?  

Line 125: Why is there a large difference in total sample size between newly emerged bees (18) and nurse bees (36)?

Line 136: What is the source of the melanin?

Was consumption rate monitored? For example, was there a difference in the amount of feed consumed between the different treatment groups?  Did bees have ad lib access to feeds?

Line 138-146: Please add further information about the statistical approach.  How were replicate cages treated?  Were they analyzed as a random factor? It seems that bees from all cages were lumped together; however, it should be demonstrated that there was no difference in cage (replicate) within treatment if they were lumped together. Include a statistical summary that includes test statistic, df, etc. beyond simply reporting the p-value. where appropriate. More detail is needed, in general, in the Results section.

Figures: Each curve represents the sum or the mean of three replicates per group? Is it possible to include a measure of variance in the curve (e.g., standard deviation, CI)?

Figure 4 : There is a small effect here and suggests that it is driven by the three extreme values for the honey diet with B-glucans. Should these be treated as outliers or what criteria were used to include them in the data analysis?

Line 208: Should include that the Felicioli study looked at response of DWV to bees given B-glucan diet. There is a missed connection here between this sentence and the previous sentence. 

Minor:

Title: The use of "wellness" may be inappropriate and suggest a change to the title.  The use of wellness has anthropomorphic connotations.  This study looked at survival for a short exposure period and the significance of the PO experiment could've have been strengthened, for example, if the bees were challenged with a pathogen and the B-glucan bees faired better than the controls.  The study shows no effect on survival and mixed effects on PO in response to B-glucans added to diet.

Line 50-51:  There is a hanging, one sentence paragraph, and suggest that it be worked into the former or trailing paragraph.

Line 54-55:  The use of invertebrate and vertebrate is sufficient, no need to include the list of specific organisms.

Line 231: 24th?  Hour?

Additional grammar and usage editing is needed.

Round 2

Reviewer 1 Report

Thanks for your clarification. No further comments.

Author Response

Authors wish to thank the Reviewer 1 for its work

Reviewer 2 Report

The authors have responded sufficiently to most of the comments from the previous review report.  However, a couple of minor points remain:

Lines 117-119: It is ok to cite Mazzei et al. as the source for choosing 0.5% concentration of B-glucans; however, what conclusion from Mazzei et al. led to this choice, e.g., bees that received 0.5% B-glucans had the greatest survival in Mazzei et al.? A few words on this justification would be useful.

Line 57-58: It appears as though only an empty line was deleted, and the sentence remains out of place. Perhaps the sentence: Schmid and colleagues [18] observed that phenoloxidase activity increased with the age of worker bees and reached a plateau within the first week of adult life COULD BE MOVED to support the choice of sampling newly-emerged and nurse bees at beginning of the Materials of Methods.

Lines 117-119: It is ok to cite Mazzei et al. as the source for choosing 0.5% concentration of B-glucans; however, what conclusion from Mazzei et al. led to this choice, e.g., bees that received 0.5% B-glucans had the greatest survival in Mazzei et al.? Including a few words on this justification would help.

Lines 150-151: My apologies for not being more clear on the source of melanin.  It is understood that melanin is the product of the reaction and this is measured by spectrophotometry.  However, how was the melanin standard prepared (e.g., a dilution of melanin)?  A bit more clarification is needed here.

Line 159-162: The additional wording on the statistical approach and test statistics is appreciated but consider adding "replicates within treatment were pooled" or similar. 

Figure Legends 1 and 2: Error bars have been added, this would seem that the values represent the average, not the sum (as stated in the legends), of the three replicates for each treatment?

Minor: Grammar and Usage

The manuscript could benefit from additional grammar and usage editing such as: Include articles (a, an, the) where appropriate; Remove possessives (s' and 's); Correction of capitalization (e.g., line 62 - Authors does not need to be capitalized). These are a couple of examples.
